# Anxiety During Employment-Seeking for Autistic Adults

**DOI:** 10.3390/brainsci15010019

**Published:** 2024-12-27

**Authors:** Tamara Hildebrandt, Kate Simpson, Dawn Adams

**Affiliations:** 1Autism Centre of Excellence, School of Education and Professional Studies, Griffith University, Brisbane 4222, Australia; t.hildebrandt@griffith.edu.au (T.H.); dawn.adams@griffith.edu.au (D.A.); 2Griffith Institute for Educational Research, Arts, Education and Law, Griffith University, Brisbane 4222, Australia

**Keywords:** employment, job-seeking, interviews, anxiety, mental health, autism, adulthood

## Abstract

Background/Objectives: Autistic adults are more likely to be unemployed compared to neurotypical adults and those with disability. To address these poorer employment outcomes, it is important to consider factors that may be impacting on autistic adults’ employment outcomes. Anxiety is a common co-occurring condition for autistic adults; however, there is little research on how anxiety affects or influences autistic people’s experience across the employment-seeking process. The aim of this study was to explore whether anxiety is perceived to affect autistic adults’ ability to engage in employment-seeking tasks. Methods: Online, semi-structured interviews were conducted with 12 autistic adults (22–52 years) who were actively seeking employment or had sought employment in the last 18 months. Interview transcripts were analysed using thematic analysis. Results: Three themes were generated from the data analysis. Theme 1, Finding the “sweet spot”, identified some level of anxiety—but not too much—was helpful in performing the employment-seeking tasks and this “spot” could vary depending on the person and the task. Theme 2, Anxiety affects the ability to perform and function, encapsulates the autistic person’s experience when anxiety is too high. The third theme, The “vicious cycle” of anxiety and employment-seeking behaviours, explores participants’ ongoing experience of anxiety on their employment-seeking behaviours. Conclusions: The findings suggest that autistic job candidates would benefit from tailored accommodations and adjustments offered throughout the recruitment process, to reduce anxiety and improve employment-seeking outcomes for autistic candidates.

## 1. Introduction

The recent rise in research focusing on autistic adults has highlighted that autistic individuals experience poorer outcomes than their neurotypical peers across a range of areas, including independent living, relationships, and employment [1,2,3,4]. In Australia, the unemployment rate for autistic adults is 18.2%, almost six times higher than for individuals without a disability and more than twice as high as for individuals with other disabilities [5]. This trend is replicated globally [6,7,8]. These poor employment trends are reported early in adulthood, with 53.4% of autistic young adults aged between 21 and 25 having never been employed [9]. Despite growing numbers of autistic young adults completing university [10], Cage and Howes [11] reported that only 33% of young autistic university graduates in the UK are employed full-time. In contrast to statistics for non-autistic people, the unemployment rate for autistic adults with graduate university qualifications is higher than for those without tertiary qualifications [12]. Identifying factors which may influence employment outcomes for autistic people is a pertinent step towards addressing these outcomes.

Transition planning [13], vocational programs [14], supportive relationships [15] and previous work experience [16] are recognised as key factors that increase young autistic adults’ prospects of gaining employment. However, there has been limited exploration of the psychosocial impact of these supports throughout the employment-seeking process. Both [13,16] noted work experience creates a ‘safe environment’ for skill development, where individuals are free from the fear of making mistakes. This brings into question whether work experience removes some of the anxiety associated with unknown work environments and employment expectations, making the process of seeking employment less anxiety triggering. Hatfield [17] noted anxiety as a moderating variable for the effectiveness of transition planning; however, the impact of anxiety during transition planning was not explored in depth.

Successful navigation of the job-seeking process is noted as a key barrier to the acquisition of employment for autistic adults [18]. In particular, employment interviews are well documented within the existing body of literature as an obstacle for successful recruitment outcomes of autistic adults [19]. Bross [15] reported that nervousness in employment interviews negatively impacted autistic young adults’ responses to interview questions, contributing to negative recruitment outcomes. Whilst the potential impact of anxiety on interview performance and outcomes was acknowledged by [15], it was not explored in depth or across other facets of the end-to-end recruitment process.

One factor that may influence employment outcomes is anxiety. Anxiety is a common co-occurring diagnosis with autism, with a reported lifetime prevalence of 42% in autistic adults [20]. Anxiety has been shown to impact autistic people’s experience at work to the point where leveraging strategies for managing stress and regulating emotions have been noted as key factors for autistic employees to be successful at work [21,22]. Young autistic adults with average or above IQs report experiencing anxiety at work that can be debilitating, and this anxiety reduces their perceived self-efficacy, personal confidence, and overall employment outcomes [23]. Djela [24] reported that autistic adults commonly experience crippling anxiety in the workplace, resulting in mental health breakdowns and terminated employment.

Although the impact of anxiety on autistic adults’ experiences in the workplace is well documented, research outlining their experience of anxiety whilst seeking employment is limited. The work that has been undertaken has tended to focus upon the impact of anxiety on the job interview. Early work in this area highlighted how anxiety is the most common challenge autistic people report experiencing in employment interviews [25]. More recent work has shown how young autistic adults are reported to experience heightened anxiety during mock [26], virtual [27] and real employment interviews [28] to the extent that it stops them from being able to show their true selves. However, there are many stages to seeking employment, both before and after the job interview, most of which have not been explored yet in the literature. This suggests that further work is needed within this area to understand more about the effect of autistic adults’ anxiety on looking for and gaining employment.

Anxiety is a subjective and complex experience and may manifest differently for an autistic person [29]. This study is positioned within an interpretivist paradigm acknowledging that how anxiety affects a person is an individual experience. To better understand this experience requires hearing the person’s account [30]. As the effect of anxiety on employment-seeking tasks for autistic people is largely unexplored, a qualitative method is considered most appropriate as it allows for in-depth exploration through semi-structured interviews. The aim of this study is furthering knowledge on whether and how anxiety affects an autistic people’s experience of seeking employment. The use of semi-structured interviews provides the opportunity to hear autistic people’s nuanced responses which may be missed when standardised measures developed for neurotypical populations are used [31]. Understanding the effect of anxiety across employment-seeking is an important step to removing potential employment barriers for autistic adults, particularly given that anxiety is treatable and preventable with effective, person-centred support. The importance of such work is highlighted through both the poor employment outcomes and the positive impact that employment can have on well-being and quality of life [32,33]. The findings from this research will provide employers, health professionals, and autistic job seekers with insight into the experiences of autistic job candidates and will highlight areas where employers could target supports or accommodations to support autistic candidates to gain employment.

## 2. Materials and Methods

### 2.1. Ethics

Ethical approval to conduct this research was obtained from the University Human Research Ethics Committee prior to the study’s commencement. This research has met all obligations and requirements outlined within the University’s human resource ethical guidelines and procedures.

### 2.2. Recruitment and Participants

The study was advertised via social media (LinkedIn and Facebook) and the University Research Newsletter. Autistic adults (formally diagnosed or self-diagnosed) who were actively seeking employment or had sought employment in the last 18 months were invited to participate. Purposeful recruitment was undertaken to include a range of participants including recent university graduates seeking employment. Once consent was provided, participants were provided a link to an online questionnaire (described below). No restrictions were placed on age or co-occurring conditions.

Twelve adults provided consent to participate in this study and completed both the online questionnaire and the interview. An overview of each participant’s demographic details and co-occurring conditions is presented in Table 1. As per the inclusion criteria, all participants were actively seeking employment or had sought employment in the last 18 months. Participants’ ages ranged from 22 to 52 years. Six (50%) of the participants identified as female, four (33%) as male, one as a transgender woman and one as non-binary gender. Ten participants had a formal autism diagnosis; the two participants who self-identified as autistic both scored above the cut off score on the AQ-10 Autism Spectrum Quotient (AQ-10) (see Section 2.3). Based on the Anxiety Scale for Autism-Adults (ASA-A) scores (see Section 2.3), nine participants were classified as having anxiety at a level that significantly impacts on their life. At the time of the interview, nine participants were currently unemployed, with seven of these actively seeking employment.

### 2.3. Descriptive and Demographic Measures

Online questionnaires were administered to gain descriptive information about the participants’ demographics (gender, age, ethnicity, employment status, diagnosis, and education), autism characteristics and level of anxiety.

The AQ-10 is a measurement tool that can be used for rapid screening of autism characteristics to support referral decisions for specialist diagnostic assessment [34]. This 10-item scale measures the degree to which respondents identify with behaviours associated with autism. Scores >6 are considered positive identifiers of autistic behaviours [34].

The Anxiety Scale for Autism-Adults (ASA-A) is a measurement scale designed to assess the level of anxiety within autistic adults, as it has strong internal consistency, convergent validity, and reliability [29]. Using this scale, participants were asked to record how often (*never, sometimes, often, always*) they had experienced each of the 20 anxiety statements over the past week. Total anxiety scores were assessed, with scores of 28 or higher indicating a participant’s life is significantly impacted by anxiety [29].

### 2.4. Procedure

After consenting and completing the online questionnaires, participants booked in their interview. Prior to the interview, participants were sent a sample of interview questions. They were also sent the Multidimensional Visual Scale for Anxiety [35], a picture-based multidimensional measure of anxiety, developed by a participatory group of scholars (including autistic scholars). The scale shows five facial images, varying from “calm” to “anxious”. Riccio et al. [35] found the scale generally interpretable, with 79.2% of participants able to interpret it correctly. Participants were informed that this scale could be used during the interview as a visual prompt to describe the levels of anxiety they may experience at different stages of the employment-seeking process. The use of visual prompts provides a concrete example of abstract concepts and can provide a more accessible interview experience for autistic adults [36].

Semi-structured interviews were conducted by the first author. An interview guide (available upon request from the authors) was prepared and reviewed by an autistic adult, who provided feedback on the wording of the questions to improve clarity. Interview questions focused on the participants’ experiences of anxiety and the impact on them throughout the employment-seeking process. Participants were asked what employment-seeking activities they had performed over the last 18 months (i.e., job search, interviews, phone calls, etc.). For each employment-seeking activity participants had engaged in, they were asked (a) their level of anxiety when performing this activity (Example question: “Thinking about the time when you were [insert employment seeking activity], how would you rate your anxiety”), (b) whether this level of anxiety impacted their performance (Example question: “How did (if at all) your anxiety impact your when you [insert employment seeking activity], and (c) whether their employment-seeking experience may differ if their anxiety level was higher or lower during this activity (Example question: “What do you think the experience would have been like if your anxiety was higher”?. Interviews were conducted using video conferencing and ranged from 30 to 80 min. Interviews were transcribed using a professional transcription service. Transcriptions were checked for accuracy against the recording and edited as required.

### 2.5. Community Involvement

The research team includes a parent of an autistic child, a parent of a neurodivergent child and a neurodivergent researcher. Combined, the team brought collective experience of working with autistic people and in human resources. Recognizing the importance of autistic input into the study, such input was sought when evaluating the relevance of the research, determining the methodology and finalizing the interview guide.

### 2.6. Data Analysis

Questionnaire data were analysed solely for the purpose of providing descriptive information about the participants and is reported under Recruitment and Participants. The data generated from the semi-structured interviews were analysed using reflexive thematic analysis [37]. This iterative, six-phase process has been used to explore the experiences of autistic adults [38,39]. An inductive approach was adopted which provided descriptive and interpretive accounts of the participants’ perspectives. The lead author immersed themselves in the dataset reflecting at the semantic level on phrases, and segments of the data. They identified areas of interest that related to the effect of anxiety on the autistic person’s experience of seeking employment. Using Microsoft OneNote windows 10, they made notes and memos to the transcript to support the development of codes, capturing consistent features existing within the data. During this period, they had regular meetings with the second author, where the codes were reviewed and discussed to ensure assumptions and interpretations of meaning associated with the codes were sound. This was an iterative process which resulted in the collapsing and merging of numerous codes, reviewing transcripts and adjusting where required. This process enhanced confirmability of the interpretation of the data. The codes and segments of data for each code were captured in separate OneNote documents and then shared with all authors. The research team discussed how codes could be grouped together based on their connectivity and relevance to the research aim. A thematic map of provisional themes and subthemes was developed. The first author checked to ensure the codes represented the dataset. The themes were reviewed, revised, and refined by the team through a process of re-engaging with the coded data and reviewed again against the entire data set. Data extracts were selected to illustrate the theme. This process of review, revision and refinement allowed the team to determine whether the qualitative data captured within the theme were internally consistent and addressed the aim of the study.

## 3. Results

Three major themes were generated from the data. Theme 1, Finding “the sweet spot”, relates to the participants’ experiences of finding the right level of anxiety (i.e., not too little and not too much) to complete the employment-seeking tasks. However, finding the sweet spot was not always possible and participants spoke about the effect on employment-seeking when anxiety levels were too high, Theme 2, Anxiety affects the ability to perform and function, was evident across different stages of the employment-seeking process. Employment-seeking was an ongoing process; the relationship between anxiety and this process was described in Theme 3, The “vicious cycle” of anxiety and employment-seeking behaviours. This vicious cycle has consequences for the autistic person’s well-being and motivation to seek employment.

### 3.1. Theme 1: Finding the “Sweet Spot”

Participants described a dichotomy in the impact of anxiety, noting it had both positive and negative impacts on well-being, performance of employment-seeking tasks, and recruitment outcomes. Some participants said that experiencing some level of anxiety was beneficial when seeking employment, noting the “sweet spot” (Geoff) or a “good balance” (Luke) between too low or too high anxiety was necessary for successful employment-seeking outcomes. When anxiety was too low, participants described feeling detached, unfocused, and lacking motivation to engage in job-seeking activities. They felt this caused them to appear inattentive, of low intelligence, and lacking in social etiquette. Deanne described how, during interviews, “If I was super calm it would mean that I had paid no attention to anything that mattered because I was just detached”. Similarly, Matthew required some level of anxiety when he was applying for jobs, claiming if he was not anxious, “I’d get bored, and I probably wouldn’t finish it. I probably wouldn’t do it”. In contrast, anxiety levels perceived as too high were deemed equally debilitating, resulting in decreased mental health and poor performance or non-performance of job-seeking activities. Importantly, the perceived anxiety sweet spot was not always at the mid-point of the anxiety scale.

Geoff suggested that having too low or no anxiety negatively impacted his ability to communicate effectively during interviews. He described presenting as unprofessional and unable to build rapport in interviews where his anxiety levels were low, explaining, “I can take the jovial, folksy, country bumpkin too far… I start to get a little bit politically incorrect”. Yet with too high anxiety during interviews, Geoff would demonstrate “clipped speech” resulting in a failure to build rapport with hiring managers. Geoff described his anxiety sweet spot for interviews as:

Probably a two and a half-ish [moderate anxiety]. That’s enough anxiety to reign my folksy, larrikin charm. Not be too anxious. Give proper answers and have a proper discussion, a professional discussion. So yeah, that two and a half probably is the sweet spot.

With this moderate level of anxiety, Geoff felt it enabled him to “actually be successful in gaining a role”.

Finding the sweet spot of anxiety also aided motivation during employment-seeking activities. Taylor explained that if her anxiety was low or she was calm, “I might just be afraid that I’d never do it. Anxiety is a bit of a motivator for me”. Similarly, Matthew described how being calm would demotivate him to search for employment, explaining, “I probably wouldn’t do it. I mean, I’d probably be less motivated… I’d be so relaxed, I’d be like ‘Oh, I’ll do it later’”. Luke similarly felt the right level of anxiety helped them prepare for interviews, stating, “I think it [the anxiety] kept me motivated to focus, to like actually look at preparing. If I was like a one or two [low anxiety] and I wasn’t worried about it… I might not have found the right resources to do it and then the interview might not have gone as well”.

Not only did some level of anxiety motivate participants, but Fernando also found it helped him focus: “Sometimes I could be doing something more intensely”, referring to tasks during the employment search and application process. Several participants felt that having some level of anxiety (low) during an interview was beneficial, enabling them to hold their “train of thought” and “think about what to say better”. Natalie felt that a moderate level of anxiety improved her ability to focus and remain attentive during interviews, explaining, “In a way maybe the anxiety helps me focus. When I’m not anxious, I do get distracted”. However, Natalie said that when she experienced high anxiety during interviews, “It’s very difficult to answer the questions”.

The sweet spot could vary depending on the employment-seeking task. Sharon explained that moderate levels of anxiety during the job search and application process allowed her to become “hyper focused” on the task. However, when she experienced moderate levels of anxiety during interviews, “[it’s] very hard to give your full attention”. For Sharon, her sweet spot anxiety level for preparing job applications was higher than her sweet spot for interviews.

### 3.2. Theme 2: Anxiety Affects the Ability to Perform and Function at Every Stage of Employment-Seeking

Participants described experiencing discomfort arising from high levels of anxiety that could be all encompassing. Some participants were able to manage their anxiety sufficiently to complete an employment-seeking task, but they were not able to maintain this state in the hours or days that followed. Deanne explained, “The aftermath is the worst part… the interview like that’s yuck, but the aftermath is the worst bit”. She went on to describe the meltdowns experienced after the interview as “debilitating” and more negative than the interview itself. For other participants, the unmanageable levels of anxiety resulted in reduced ability to function and perform employment-seeking tasks, which often led to poorer performance and employment outcomes. In some cases, unmanageable levels of anxiety resulted in participants having to withdraw from or abandon the employment-seeking at that point in time. Participants reported that this was evident across a range of employment-seeking tasks.

During the job application process, Luke found when he experienced moderate to high levels of anxiety, he was more indecisive, and it took him longer to complete tasks. When applying for jobs, he stated, “It’s hard for me to focus. It’s hard for me to shift my focus… making me hesitant… So that I spend a lot of time just deliberating”. He went on to say, “I might rewrite a sentence like 10 times just to see if I’m happy enough with the wording”. In other cases, when the person’s anxiety was too high, they would abandon the job application process. Fernando reported that with moderate to high levels of anxiety “it’s too much” and he had to abandon tasks and “step away from the situation”. Laura also recalled having to abandon the task of preparing a job application, reflecting: “It was so bad, and I really actually panicked so much that I walked out, and I was in my office, and I was almost hyperventilating”.

Once a person was successful in obtaining an interview, anxiety could impact on their ability to prepare for it. Laura described how high levels of anxiety would reduce her ability to function when preparing for job interviews; she would need to “read something three times” for it to “sink in”. Anxiety could reach a point where the person was unable to complete the preparation at that time. Jen recalled experiencing high anxiety when preparing for a job interview which caused her to walk outside and have a meltdown in her driveway, explaining, “I was so stressed out that I stood in my driveway and cried for 10 min. I was freaking out”. For other participants, this might involve having to postpone the interview. Geoff explained how he would “feign illness or reschedule an appointment” if he could not drag himself out of “that really toxic level” of anxiety.

A high level of anxiety was commonly reported as detrimental to the person’s ability to function during the job interview. Anthony felt that being more anxious would impair his performance in job interviews, stating, “I’d come across as less confident and fixate on certain things and not be able to bring good examples to mind easily”. Rose described how she was able to perform in practice interviews, saying, “I’m able to think about them [questions]. I’m able to think through the problems that may arise and plan a strategy”. She went on to explain how this was different in real-life interviews where her anxiety levels were high: “Unless I can get myself back down. I can’t perform it. It’s really upsetting, confronting and scary, because if I can’t perform an actual interview how am I going to get an actual job”?

For some participants, the effort in trying to manage their anxiety during the interview impacted on their performance. Laura described how the effort of trying to “create fake calm” impacted on her interview performance, saying, “I might look calm to other people, but then my answers might be garbage because I’m so busy focusing on being calm”. Similarly, Sharon described anxiety during interviews as being all consuming, where her focus would be directed to managing her anxiety rather than focusing on performing during the interview:

I’d be a wreck really and really just trying to keep it together through the interview more than actually being present in the interview… I’ll be, you know… getting my anxiety level up and I’ll be sweating like crazy. It’s just harder to focus my mind in one spot. I’ve learned to sort of ask questions to give myself time to think. But as I said, through that period that nothing really helped me at all.

For some employment positions, the interview process included the interviewee having to participate in a range of tests. For Geoff, psychometric testing triggered his performance anxiety, which in turn caused fatigue and impaired cognition. He said, “Being anxious about not being good enough, I get fatigued really easily when I concentrate. So… I have the mental capacity of a cabbage”.

### 3.3. Theme 3: The “Vicious Cycle” of Anxiety and Employment-Seeking Behaviours

The influence of anxiety on the employment-seeking process was described by many participants as a “vicious cycle”. The cycle commenced with anxiety negatively impacting performance in employment-seeking activities, which led to unsuccessful recruitment outcomes. This in turn increased participants’ anxiousness about participating in future employment-seeking activities, and subsequently impacted participants’ mental health and their ongoing ability to acquire employment. Sharon said,

And then obviously the confidence level every time you completely mess it up adds to that anxiety for the next time round, ’cause you are just feeling that little bit more pressure and disappointment I guess in yourself would be as to “why, why does everybody else seems to be able to do this”?

Laura described how the continuous rejections to job applications heightened her anxiety, rendering her debilitated when undertaking employment-seeking tasks where she found even the “simplest tasks really difficult”.

The ongoing anxiety associated with employment-seeking activities and the subsequent unsuccessful outcomes “would inevitably lead to another downward spiral”, reported Taylor; it left her wondering, “Why am I even bothering”?. Geoff also described the “cumulative” effect of anxiety throughout the employment-seeking process, where the rejection from participating in previous anxiety-ridden job search activities only decreased his enthusiasm to apply for roles. Sharon noted the impact of this vicious cycle on her mental health, explaining, “The confidence level every time you completely mess it up adds to that anxiety for the next time round”. Deanne described great difficulties from dealing with ongoing rejection and anxiety when seeking employment:

The anxiety that we’re not going to be good enough, and we have to go through it again and again and again, and continually face rejection simply because we didn’t look them in the eye enough or we might be fiddling with our hands.

For some participants, the vicious cycle of anxiety and unsuccessful employment outcomes contributed to long-term negative mental health and well-being. Participants spoke of a variety of impacts on their well-being, including self-doubt, decline in their self-esteem, feeling sad or frustrated, to feeling depressed with suicidal ideations. For many, disengaging from employment-seeking was a proactive strategy to avoid the ongoing psychological harm resulting from engaging in an anxiety-ridden employment-seeking process. Anthony described the impact of his emotional state on his ability to engage in job-seeking activities, explaining, “After a while I just couldn’t keep it up because I was so depressed, despondent, annoyed. Fed up with it”. He noted that if he had felt a sense of calm throughout the employment-seeking process, he may have “continue[d] looking for work without making my [his] life unbearable”. Concern about the long-term mental health impact that seeking employment had on themselves and on the broader autistic community was expressed by many participants. As one explained, “We are suffering really badly and we’re really talented, capable people and we are dealing with unnecessary suicidal ideation and mental health impacts”.

## 4. Discussion

This is the first study to specifically explore the role of anxiety on a range of employment-seeking tasks for autistic people. This expands upon previous work which has tended to focus only upon anxiety experienced during interviews. This work highlights the consistent and complex nature of anxiety which impacts autistic people across the entire employment-seeking process. Participants spoke about needing to attain their anxiety “sweet spot” to be able to undertake and be successful at the employment process. However, this sweet spot varied across the different employment-seeking tasks (e.g., applications, interviews) and also varied across individuals. As finding this sweet spot was not always achievable, participants also described how anxiety impacted them across the entire range of employment-seeking tasks, resulting in reduced performance. The influence of anxiety on employment-seeking was ongoing and resulted in participants feeling like they were caught in a vicious downward cycle which had an impact on the autistic person’s mental health and well-being and their motivation to seek employment and, therefore, their employment success.

Employment-seeking is a goal-directed process which requires high levels of self-regulation to direct cognition, affect and behaviour, often over extended periods of time [40]. Autistic adults spoke about this self-regulation in terms of ensuring an anxiety level that motivated them to get the task done but did not impact their outcomes—the anxiety “sweet spot”. This sweet spot is dynamic (based on the task) and variable (based on the person themselves). This finding extends the existing literature, which has tended to focus on negative aspects of anxiety for autistic adults [20,41]. The anxiety sweet spot could be interpreted in relation to Yerkes–Dodson law [42] which denotes that the relationship between task performance and anxious arousal for neurotypical individuals is normally distributed, where optimal performance is achieved for some tasks at moderate levels of anxiety, with both low and high levels of anxious arousal correlating with weaker task performance [42,43]. The Yerkes–Dodson law has not been explored specifically in autistic people, although the results of this work suggest that the relationship between optimal anxiety and task performance for autistic job seekers may be far more complex than suggested in this law. Given that autistic adults have a higher baseline level of arousal than their neurotypical peers, living and coping with anxiety as a normal experience [44,45], it is understandable that some level of anxiety may be required for optimal job-seeking task performance. However, the finding that too low anxiety could negatively impact an autistic individual’s performance throughout the employment-seeking process due to poor communication, low concentration, or lack of motivation to present oneself positively is noteworthy, and one that may benefit from further exploration.

When regulation of their anxiety level was not possible, participants described an impaired ability to function across employment-seeking tasks (including impaired cognition), which is consistent with other studies [43,46]. Sung et al. [23] found that young autistic adults aged between 18 and 25 experienced anxious arousal to the point where it was debilitating and damaging for employment outcomes. This present study demonstrated this is also true across a range of job-seeking contexts and that the impact spans further across the lifespan, between the ages of 22 and 55 years. Heightened anxiety levels were noted as impacting participants’ ability to communicate and respond to questions, causing them to withdraw from employment-seeking activities including interviews; this is consistent with existing research [15,45]. Anxiety may also amplify executive functioning challenges, hindering a person’s ability to focus on questions asked in interviews or to recall relevant work experiences and to structure these in an appropriate response [47,48]. This brings into question the use of behavioural event interviews for autistic candidates, which require interviewees to retrieve relevant work experiences from long-term memory to formulate responses. These are highly demanding cognitively for neurotypical adults [49] so would be even more demanding for autistic adults in a typical interview setting [50,51,52]. Indeed, the Buckland review in the UK made a recommendation to “modernize” the recruitment process, stating that “the interview may not be a fair way or necessary way to recruit autistic candidates” [53] (p. 29).

Several participants noted the negative long-term mental health impact of continuous engagement in anxiety-inducing employment-seeking activities, with participants feeling disparaged, depressed and reluctant to engage in further employment-seeking. This finding is important, as research to date has focused on the relationship between unemployment or underemployment of autistic adults and poor mental health or quality of life outcomes [54,55]. However, these findings provide insights into how the actual act of seeking employment can itself have damaging repercussions on mental health for autistic adults, and that the mental health challenges which stem from employment-seeking then become a barrier to employment-seeking (or successful employment-seeking). In non-autistic adults, employment is seen as a protective factor against mental health [56], so this potentially differing relationship warrants further exploration and attention.

### 4.1. Limitations

The participants in the current study were recruited purposively so that their shared experiences could be analysed. The sample size within this research is consistent with other qualitative studies; however, the results of the current study may not apply to autistic adults with different demographic and behavioural characteristics who are searching for employment. As qualitative research methods were applied, conclusions regarding causal relationships for anxiety during employment transitions cannot be made. The interviews and data collection took place in 2021 when many Australian states and territories were experiencing COVID-19 restrictions. In instances where participants reflected upon their experiences of anxiety during employment-seeking activities that were undertaken in 2020 and 2021, these activities may have been delivered in an adjusted manner to ensure compliance with COVID-19 restrictions (e.g., using virtual interviews or onboarding meetings, etc.). Whilst this may not have significantly impacted on the results, lower employment opportunities during this timeframe contributed to the decision to use a smaller sample size.

### 4.2. Future Directions

Because this was an initial study in this area, the decision was made to focus on the impact of anxiety in employment-seeking in autistic people in general. Therefore, the interview questions did not enquire about specific anxiety subtypes and their impact on specific tasks. Ambrose [57] discussed potential pathways for anxiety to interact with autism characteristics and impact academic and social outcomes for autistic people which could inform this work. For example, expanding upon Spence and Rapee’s [58] model of social anxiety, they suggest that social rejection and perceived failure in social situations may lead to negative self-perceptions, which may increase the likelihood of social anxiety in that situation in the future. Participants in this study discussed times that they had felt rejected socially or that they had failed at parts of job-seeking that were dependent upon specific social outcomes or interactions, suggesting that this model may be useful to draw upon. This more detailed, nuanced approach could be the focus of future research.

When asked about the impact of anxiety on employment-seeking tasks, participants also spoke about the mental health impact of seeking employment, suggesting a cyclical relationship between the two. Whilst the existing body of research has focused on the relationship between unemployment or underemployment of autistic adults and poor mental health or quality of life outcomes [54,55], this study suggests that continuous engagement in an anxiety-inducing, employment-seeking process with unsuccessful outcomes could also contribute to poor mental health and quality of life for autistic adults. This is therefore a critical area for future research, and one that should be explored in collaboration with the autistic community.

The literature to date on workplace or employer accommodations has focused on adjustments required for autistic adults to maintain employment [59,60], whereas the findings from this study support the recommendations from Maras et al. [61], suggesting that there is an opportunity for employers to review the accommodations offered within the recruitment process itself, to support the acquisition of employment for autistic adults. It may be that gathering evidence on the impact of these accommodations will provide greater support for their implementation in recruitment settings.

## 5. Conclusions

The results indicate that whilst suboptimal levels of anxiety can have negative effects on autistic adults during the employment-seeking process, optimal levels of anxiety can have positive impacts, particularly for attention and motivation. This optimal, “sweet-spot” level of anxiety varies for individuals and for employment-seeking tasks, highlighting the importance of listening to individuals to hear their unique experiences. Further research is required to understand the types of supports that would be beneficial for autistic adults who experience non-optimal levels of anxiety throughout the employment-seeking process.

## Figures and Tables

**Table 1 brainsci-15-00019-t001:** Participant demographic information.

Pseudonym	Age	Gender	Ethnicity	Highest Education	Employment Status	Co-Occurring Conditions	Autism Diagnosis	AQ-10 Score	ASA-A Score
Taylor	25	Transgender woman	White	Bachelor’s degree (currently studying honours)	Unemployed, not currently seeking employment/Previously unsuccessful at seeking employment/Studying	Anxiety, depression	Formally diagnosed	7	39
Jen	23	Female	Australian/White	Graduate Diploma	Unemployed—seeking employment	Anxiety, depression, coordination/motor difficulties, e.g., dyspraxia	Formally diagnosed	8	41
Laura	39	Female	Bosnian	Graduate Diploma (currently studying Doctoral degree)	Unemployed—seeking employment/previously unsuccessful at seeking employment	ADHD, ADD, ODD	Formally diagnosed	7	40
Luke	22	Non-binary	Australian/European	Currently studying Bachelor’s degree	Unemployed—seeking employment/studying (full-time or part-time)	Anxiety, depression	Informally diagnosed	10	33
Matthew	32	Male	White	Bachelor’s degree (currently studying Graduate Certificate)	Unemployed—seeking employment/studying (full-time or part-time)	None	Formally diagnosed	7	24
Deanne	37	Female	Australian/White	Bachelor’s degree (currently studying)	Paid employment—full-time or part-time/studying (full-time or part-time)	ADHD/ADD/ODD, anxiety, depression, coordination/motor difficulties, e.g., dyspraxia, CPTSD	Formally diagnosed	8	30
Geoff	49	Male	European/White	Bachelor’s degree	Unemployed—seeking employment/previously unsuccessful at seeking employment	ADHD/ADD/ODD, anxiety, depression	Formally diagnosed	7	37
Rose	43	Female	European/Maori	Graduate Diploma	Unemployed—seeking employment/previously unsuccessful at seeking employment/unpaid employment-volunteer (full-time or part-time)	IDD, anxiety, depression, coordination/motor difficulties, e.g., dyspraxia, PTSD, OCD, BPD, anorexia, bulimia, schizophrenia	Formally diagnosed	10	42
Sharon	52	Female	Australian	Bachelor’s degree (currently studying Graduate Certificate)	Paid employment—full-time or part-time/studying (full-time or part-time)	ADHD/ADD/ODD, anxiety, depression	Informally diagnosed	10	28
Natalie	46	Female	Australian	Graduate Diploma (currently studying Master’s degree Coursework)	Unemployed—not seeking employment/studying (full-time or part-time)	Anxiety, depression	Formally diagnosed	7	39
Anthony	39	Male	Anglo-Celtic/White	Graduate Diploma	Paid employment—full-time or part-time	None	Formally diagnosed	5	6
Fernando	36	Male	Sri Lankan	Bachelor’s degree	Unemployed—seeking employment	Anxiety, depression	Formally diagnosed	8	20

ADHD = attention deficit hyperactivity disorder; ADD = attention deficit disorder; OCD = obsessive compulsive disorder; ODD = oppositional defiance disorder; PTSD = posttraumatic stress disorder; CPTSD = complex post-traumatic stress disorder.

## Data Availability

The original contributions presented in this study are included in the article. Restrictions apply to the availability of the original data records due to ethical reasons.

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
