# Peer review of "Anxiety During Employment-Seeking for Autistic Adults"

_brainsci, 2024, doi:10.3390/brainsci15010019_

Round 1
Reviewer 1 Report
Comments and Suggestions for Authors
Dear Editors and Authors,
Thank you for inviting me to review the manuscript "Anxiety During Employment-Seeking for Autistic Adults." This qualitative study investigates how anxiety affects autistic adults' experiences during job seeking. While the topic is both relevant and valuable, addressing an important gap in our understanding of employment challenges faced by autistic adults, I have several suggestions for improvement that would strengthen the manuscript's contribution to the field.
The abstract requires substantial revision as it currently lacks a clearly stated research objective. Additionally, the phrase "To address this issue" lacks clear antecedent context and should be specified to indicate which issue is being addressed. The introduction would benefit from a more thorough literature review that establishes the significance of this research within the broader context of autism and employment studies.
The conceptual framework would benefit from a more precise definition of the type(s) of anxiety being investigated, along with a theoretical framework explaining how different forms of anxiety may impact employment-seeking behavior. This framework should draw from both anxiety research and autism literature to create a comprehensive foundation for the study.
Regarding methodology, the rationale for choosing a qualitative approach should be better explained, including the advantages of qualitative methodology over quantitative approaches for addressing the research questions. The authors should discuss how their methodological choice aligns with both the research objectives and the specific needs of autistic participants. In terms of measures, full titles should be provided before using abbreviations (AQ-10 and ASA-A). The "Multidimensional Visual Scale for Anxiety" should be integrated into the measures section rather than being presented separately. Consider consolidating all measurement instruments into a single, comprehensive section, with detailed explanations of how each instrument contributes to understanding the research questions.
A significant concern regarding data collection is the interview question "whether or how their employment-seeking experience may differ if their anxiety level was higher or lower during this activity," which may be challenging for autistic participants to answer. This question might benefit from revision to be more concrete and accessible, perhaps by breaking it down into more specific, situation-based questions. Furthermore, the data analysis section requires more detailed explanation of the procedure for theme extraction, including specific analytical methods used to identify and validate the three themes from the interview data. The authors should describe their coding process, including how they ensured reliability and validity in their qualitative analysis.
Author Response
We have uploaded a document with our response to the comments made by reviewer one.

Reviewer 2 Report
Comments and Suggestions for Authors
The paper is quite fluent and well-organized; the hypothesis aligns with the results and conclusions. I have a few minor suggestions.
First, it would be important to specify in the methodology whether the authors used specific software for thematic analysis. Additionally, providing a concise output of the prevalent themes, for example in the form of a summary table, would enhance clarity.
Furthermore, the role of stigma towards individuals with autism and mental health disorders in general could be briefly discussed, particularly regarding how stigma may hinder employment opportunities, as highlighted in a recent paper: “The link between stigmatization, mental health, disability, and quality of life” https://ampphealthjournal-network.org/wp-content/uploads/2024/11/AMPPH_2025_254-260.pdf. Employment, in turn, could be considered a protective factor for individuals with such diagnoses.
Lastly, the paper could briefly propose programs for anxiety reduction in this specific population, citing experimental studies on the topic (e.g., Hillier, A.J., Fish, T., Siegel, J.H., et al. Social and Vocational Skills Training Reduces Self-reported Anxiety and Depression Among Young Adults on the Autism Spectrum. Journal of Developmental and Physical Disabilities, 23, 267–276 (2011). https://doi.org/10.1007/s10882-011-9226-4).
Kind Regards
Author Response
We have responded to the reviewer's comments in the attached fie.

Round 2
Reviewer 1 Report
Comments and Suggestions for Authors
The authors have addressed my comments.
Comments on the Quality of English LanguageI can understand the english expression in the manuscript.